# Association of Phosphorylated Pyruvate Dehydrogenase with Pyruvate Kinase M2 Promotes PKM2 Stability in Response to Insulin

**DOI:** 10.3390/ijms241813697

**Published:** 2023-09-05

**Authors:** Abu Jubayer Hossain, Rokibul Islam, Jong-Bok Seo, Hwee-Seon Park, Jong-Il Kim, Vikas Kumar, Keun Woo Lee, Jae-Bong Park

**Affiliations:** 1Department of Biochemistry, Hallym University College of Medicine, Chuncheon 24252, Republic of Korea; yubayerbge89@gmail.com (A.J.H.); rakibbgeiu@yahoo.com (R.I.); 2Institute of Cell Differentiation and Aging, Hallym University, Chuncheon 24252, Republic of Korea; 3Department of Biotechnology and Genetic Engineering, Faculty of Biological Science, Islamic University, Kushtia 7003, Bangladesh; 4Korea Basic Science Institute Seoul Center, Anamro 145, Seongbuk-gu, Seoul 02841, Republic of Korea; sjb@kbsi.re.kr; 5Department of Biomedical Sciences, Genomic Medicine Institute, Medical Research Center, Seoul National University College of Medicine, Seoul 03080, Republic of Korea; hwee38@snu.ac.kr (H.-S.P.); jongil@snu.ac.kr (J.-I.K.); 6Division of Life Science, Department of Bio and Medical Big-Data (BK4 Program), Research Institute of Natural Science (RINS), Gyeongsang National University (GNU), 501 Jinju-daero, Jinju 52828, Republic of Korea; vikaspathania777@gmail.com (V.K.); kuku1004@gmail.com (K.W.L.); 7Angel i-Drug Design (AiDD), 33-3 Jinyangho-ro 44, Jinju 52650, Republic of Korea

**Keywords:** p-PDH, PKM2, insulin, protein stability

## Abstract

Insulin is a crucial signalling molecule that primarily functions to reduce blood glucose levels through cellular uptake of glucose. In addition to its role in glucose homeostasis, insulin has been shown to regulate cell proliferation. Specifically, insulin enhances the phosphorylation of pyruvate dehydrogenase E1α (PDHA1) at the Ser293 residue and promotes the proliferation of HepG2 hepatocellular carcinoma cells. Furthermore, we previously observed that p-Ser293 PDHA1 bound with pyruvate kinase M2 (PKM2) as confirmed by coimmunoprecipitation. In this study, we used an in silico analysis to predict the structural conformation of the two binding proteins. However, the function of the protein complex remained unclear. To investigate further, we treated cells with si-PDHA1 and si-PKM2, which led to a reduction in PKM2 and p-Ser293 PDHA1 levels, respectively. Additionally, we found that the PDHA S293A dephospho-mimic reduced PKM2 levels and its associated enzyme activity. Treatment with MG132 and leupeptin impeded the PDHA1 S293A-mediated PKM2 reduction. These results suggest that the association between p-PDHA1 and PKM2 promotes their stability and protects them from protein degradation. Of interest, we observed that p-PDHA1 and PKM2 were localized in the nucleus in liver cancer patients. Under insulin stimulation, the knockdown of both PDHA1 and PKM2 led to a reduction in the expression of common genes, including KDMB1. These findings suggest that p-PDHA1 and PKM2 play a regulatory role in these proteins’ expression and induce tumorigenesis in response to insulin.

## 1. Introduction

Insulin is a crucial signalling molecule, which has a variety of physiological functions including glucose uptake and glycogen synthesis, as well as the biosynthesis of proteins and lipids in cells. Additionally, insulin promotes cell proliferation and increased insulin, in addition to insulin-like growth factor 1 (IGF1) and IGF2 levels, critically stimulates tumour initiation and progression in insulin-resistant patients [1]. Insulin receptor is composed of two subunits, including α and β subunits, and has a tyrosine kinase activity for autophosphorylation. The insulin signalling pathway involves the binding of insulin to its receptor, which subsequently transmits its signal to downstream components, including insulin receptor substrate (IRS), phosphatidyl inositol-3-kinase (PI3K), Akt/PKB, glycogen synthase kinase-3β (GSK-3β), and mTOR [2,3]. Notably, Akt phosphorylates Ser9 of GSK3β (p-Ser9 GSK-3β), which converts GSK3β to an inactive form, leading to the activation of glycogen synthase (GS) [4,5]. Notably, insulin receptor (IR) exists as two isoforms: isoform A (IR-A), which lacks exon 11, and isoform B (IR-B), which includes exon 11. IR-A and IR-B bind to native insulin with a similar affinity, while the binding affinity of IR-A for IGF1 and IGF2 is significantly higher than that of IR-B [1]. Moreover, IR is overexpressed in cancer cells, and IR-A, together with the autocrine production of its ligand IGF2, is a feature of several malignancies [6].

Pyruvate dehydrogenase (PDH) complex (PDC) in mitochondria oxidizes pyruvate to produce acetyl-CoA plus CO_2_. The acetyl-CoA then enters the tricarboxylic acid (TCA) cycle, leading to ATP generation in the mitochondria matrix. PDC is composed of PDHE1, dihydrolipoyl transacetylase E2, and dihydrolipoyl dehydrogenase E3 subunits [7]. The phosphorylation of the PDH E1α subunit (PDHA1) at the Ser293 residue in mouse (Ser264 residue in human) attenuates the PDC enzyme activity, which is a major regulatory mechanism. Insulin stimulates the phosphorylation of PDHA1 at the Ser293 residue, leading to a reduced PDHA1 enzyme activity in HepG2 hepatocellular carcinoma cells. However, insulin attenuates p-PDHA1 levels in normal liver tissue and hepatocyte [8,9].

Pyruvate kinase (PK) catalyses the converting phosphoenol pyruvate (PEP) plus ADP to pyruvate plus ATP. Four PK isoforms exist, including PKL, PKR, PKM1, and PKM2. PKM1 and PKM2 are alternatively spliced and expressed from the PKM1/2 common gene. PKM1 is typically expressed in normal adult tissues, while PKM2 is highly expressed in embryo and cancer tissues [10]. PKM2 has an additional function of protein kinase activity. Upon EGF receptor’s activation, PKM2 phosphorylates Thr11 of histone 3 (H3) and induces cyclin D1 and c-Myc expression with tumour cell proliferation [11]. PKM2 also phosphorylates Stat3 at the Tyr705 residue [12]. Furthermore, a high expression of PKM2 is associated with poor prognosis in hepatocellular carcinoma (HCC) patients [13].

In a recent study, it was reported that insulin induced the phosphorylation of PDHA1 at Ser293 in HepG2 hepatocellular carcinoma cells through the RhoA/ROCK/GSK-3β signalling pathway [8]. Additionally, it was found that p-PDHA1 directly bound to PKM2, forming a complex that could translocate to the nucleus and regulate the expression of specific genes, such as LINC00273, through histone acetylation [9]. In the present study, we aimed to investigate the effect of p-PDHA1 and PKM2 on their partner proteins and discover whether p-PDHA1 can stabilize PKM2 in response to insulin.

## 2. Results

### 2.1. p-PDHA1 Interacts with PKM2

In the previous report, we reported that insulin increased p-ser293 PDHA1 and PKM2 levels in HepG2 and Huh7 hepatocellular carcinoma cells. Interestingly, we found that p-Ser293 PDHA1 formed a complex with PKM2, which translocated to the nucleus and regulated the expression of specific genes such as LINC00273 [9]. In this study, we aimed to investigate the interaction between p-Ser293 PDHA1 in mouse (p-Ser264 in human) and PKM2. We used an in silico prediction to determine the 3D structure of the protein complex and identified the amino acid residues involved in the interaction (Figure 1A,C). Specifically, we performed a comparison of the configurations of p-Ser264 PDHA1 (shown in grey, in human) and Ser264 PDHA1 (shown in pink, in human) and observed a significant disparity between them (Figure 1B). We used the RMSF (root-mean-square fluctuation) to identify the amino acid residues that contributed the most to the molecular motion and found that the region around Ser264/p-Ser264 showed a significant difference (Figure 1D). Furthermore, we used the RMSD (root-mean-square deviation), which is a numeric measurement that represents the difference between two structures, to reveal a distinct difference in p-Ser264 PDHA1 (Figure 1E).

### 2.2. DCA Reduces PKM2 Levels

To confirm that IR-A is involved in insulin signalling for p-PDHA and PKM2 expression regulation, we transfected si-IR-A to HepG2 cells. As expected, si-IR-A reduced p-PDHA1 and PKM2 expression in HepG2 cell and Huh7 cells (Figure 2A,B, respectively). Insulin treatment increased p-PDHA1 and PKM2 levels in HepG2 and Huh7 cells in a time-dependent manner. Importantly, we found that the treatment of DCA (dichloroacetic acid), an inhibitor PDH kinase, reduced both p-Ser293 Ser PDHA1 and PKM2 levels, suggesting that p-PDHA1 induced by insulin prevents PKM2 from being reduced (Figure 2C,D). Notably, DCA treatment also prevented the insulin-stimulated proliferation of HepG2 and Huh7 cells, as shown in Figure 2E,F, respectively.

### 2.3. Effects of p-PDHA1 on PKM2

Treatment with insulin increased PKM2 expression, but the knockdown of PDHA1 using siRNA prevented PKM2 expression, as shown in Figure 3A. We found that si-PDHA1 reduced PKM2 protein levels and reconstituted PDHA1 WT and the PDHA1 S293D phospho-mimetic form but not the PDHA1 S293A dephospho-mimetic form, rescued PKM2 protein levels (Figure 3B). In the presence of cyclohexamide (CHX), an inhibitor of protein synthesis, si-PDHA1 reduced PKM2 levels in a time-dependent manner (Figure 3C). However, the administration of MG132, a proteasome inhibitor, and leupeptin, an inhibitor of proteases present within lysosomes, effectively prevented the reduction in PKM2, even in the presence of the PDHA1 S293A dephospho-mimetic form. These results suggest that the reduction in PKM2 caused by PDHA1 S293A is due to protein degradation, as shown in Figure 3D. Accordingly, si-PDHA1 reduced PKM2 enzyme activity and reconstituted PDHA1 WT and the S293D phospho-mimetic form but not the PDHA1 S293A dephospho-mimetic form, recovered the PKM2 enzyme activity (Figure 3E). Notably, si-PDHA1 reduced cell proliferation and reconstituted PDHA1 WT and the S293D phospho-mimetic form but not the PDHA1 S293A dephospho-mimetic form, rescued cell proliferation (Figure 3F).

### 2.4. Effects of PKM2 on p-PDHA1

In HepG2 cells, insulin treatment led to an increase in p-PDHA1 and PKM2 levels. Our results showed that knocking down PKM2 using siRNA decreased the levels of p-Ser293 PDHA1 similarly to the knockdown of PDHA1 (Figure 4A). We could assume that PDH enzyme activity increased as p-PDHA1 levels were reduced despite constant PDHA1 levels in the presence of si-PKM2 (Figure 4A). However, si-PKM2 attenuated PDH enzyme activity, suggesting that PKM2 binding to p-PDHA1 may regulate PDHA1 activity. Moreover, insulin treatment led to a decrease in PDH enzyme activity, possibly through the Ser293 phosphorylation of PDHA1 (Figure 4B). In addition, si-PKM2 treatment resulted in a significant reduction in p-PDHA1 levels in the presence of cyclohexamide (CHX), an inhibitor of protein synthesis (Figure 4C). These results suggested that PKM2 regulated PDHA1 enzyme activity. Hence, we explored the possibility of PKM2 functioning as a protein kinase to phosphorylate PDHA1 and regulate PDHA1 activity. To test this hypothesis, we treated a purified recombinant PDHA1 protein, expressed in *E. coli*, with PKM2 and ATP or PEP. We excluded contaminated PDHK action using DCA, a PDHK inhibitor. As the recombinant PDHA1 purified from *E. coli* was already phosphorylated, we treated beads-GST-PDHA1 with alkaline phosphatase (ALP) to remove phosphate from the protein (Figure 4D,E). Our findings revealed, contrary to our initial hypothesis, that PKM2 did not phosphorylate PDHA1 with ATP or PEP, but PDHA1 itself was phosphorylated in the presence of ATP or PEP (Figure 4D,E, respectively). These results led us to conclude that PKM2 regulated PDHA1 enzyme activity through unidentified mechanisms within the protein complex, rather than through direct phosphorylation of PDHA1. Additionally, our results showed that the knockdown of PDHA1 with siRNA led to a reduction of PKM2 levels (Figure 3A). As K305 acetylation in PKM2 was reported to decrease PKM2 enzyme activity and to promote its lysosomal dependent degradation [14], we tested acetylated the stability of PKM2 at Lys305 in the absence of pPHDA1. We observed that reconstituted PKM2 WT and the PKM2 K305R deacetylated mimic form recovered PKM2 levels, whereas the PKM2 K305Q acetylated mimic form reduced PKM2 levels compared to the WT and K305R forms in si-PDHA1 transfected HepG2 cells. This suggests that acetylated PKM2 at the Lys305 residue is unstable in the absence of PDHA1 (Figure 4F). Indeed, p-PDHA1 did not coimmunoprecipitate with PKM2 K305Q, indicating that there was no association between p-PDHA1 and acetylated PKM2 at the Lys305 residue. This lack of association results in the instability of PKM2 (Figure 4G). Furthermore, si-PKM2 treatment led to a significant reduction in cell proliferation (Figure 4H).

### 2.5. Nuclear Localization of p-PDHA1 and PKM2 in Human Hepatocellular Carcinoma Cells and Tissues

Insulin was found to upregulate RhoA-GTP levels in HepG2 and Huh7 cells, but no significant change was observed in RhoA-GTP levels in rat normal liver tissue (Figure 5A). Additionally, insulin increased p-PDHA1 levels and its coimmunoprecipitation with PKM2. The use of Tat-C3 Rho inhibitor and Y27632 ROCK inhibitor reduced p-PDHA1 levels and coimmunoprecipitation with PKM2, indicating the potential regulatory role of RhoA and ROCK in PDHA1 phosphorylation (Figure 5B). Insulin also led to the upregulation of PKM2 and p-PDHA1 levels and their nuclear localization, which was prevented by Tat-C3 and Y27632, particularly in case of p-PDHA1 (Figure 5C). We further investigated the levels of p-PDHA1 and PKM2 in liver tissue from patients with human hepatocellular carcinoma (HCC), hepatoblastoma (HB), and cholangiocarcinoma (ChC). The localization of p-PDHA1 and PKM2 in HCC, HB, and ChC tissues were significantly higher in the nucleus compared to the control non-neoplastic hepatocyte tissues around tumour tissues, suggesting that nuclear localization of p-PDHA1 and PKM2 may be a characteristic feature of liver cancer tissue (Figure 5D–F). Furthermore, the number of cells was found to be increased in tumour tissues of HCC, HB, and ChC (Figure 5G–I).

### 2.6. Target Genes of p-PDHA1 and PKM2

As hypothesized, we investigated the potential regulatory role of the p-PDHA1/PKM2 complex in gene expression by a transcriptome analysis. We found that insulin induced the expression of many genes, and that si-PDHA1 plus reconstituted PDHA1 S293A markedly reduced gene expression. However, we were unable to determine whether these effects were direct or indirect. To address this, we performed chromatin immunoprecipitation (ChIP) sequencing using p-PDHA1 and PKM2 antibodies in insulin-stimulated HepG2 cells. We identified a group of genes whose promoters were associated with p-PDHA1 and/or PKM2 with an enrichment fold above 4.5. Among them, we selectively chose the target genes that were both interesting and relevant to our current ongoing projects in the laboratory. Specifically, we found that the promoters of GPR174 (<1 kb) and KDM1B (<1 kb) were identified to be associated with p-PDHA1 and PKM2, while the distal intergenic region (−62 kb) of IL13RA2 was associated with PKM2 (Figure 6A). We found that insulin increased KDM1B, GPR174, and IL-13Rα2 expression in HepG2 and Huh7 cells (Figure 6B,C, respectively). In addition, si-PKM2 reduced GPR174 and KDMB1 in HepG2 and Huh7 cells (Figure 6D,E, respectively), and si-PDHA1 reduced GPR174 and KDMB1 in HepG2 cells (Figure 6F).

Of note, ChIP-PCR experiments revealed that PKM2 and p-PDHA1 bound to the promoter region of GPR174 and KDM1B genes (Figure 6G,H, respectively), suggesting that PKM2 and PDHA1 regulate the expression of these genes through binding to their promoters. However, we did not observe an association of p-PDHA1 and /or PKM2 with the IL13RA2 promoter. This is because p-PDHA1 and/or PKM2 were expected to bind to the distal intergenic region, suggesting that insulin induces the expression of IL13Ra2 through a pathway unrelated to the p-PDHA1/PKM2 complex. Additionally, we investigated the effect of these genes on cell proliferation. Si-KDMB1 significantly reduced HepG2 cell proliferation, while si-GPR174 and si-IL13RA2 did not demonstrate a significant effect (Figure 6I,J, respectively). Importantly, we observed that higher levels of KDM1B transcript were associated with poorer survivability in patients with HCC (Figure 6K), while patients with kidney renal papillary cell cancer with a higher expression of GRP174 also showed poorer survivability (Figure 6L). These findings suggest that PKM2 and p-PDHA1 play important roles in regulating the expression of KDM1B genes, which may have implications for the development and progression of certain cancers.

## 3. Discussion

In this study, we provided evidence that phosphorylated Ser293 PDHA1 (p-Ser293 PDHA1) protects and stabilizes PKM2 from protein degradation, as shown in Figure 2 and Figure 3. Our results indicate that p-PDHA1 and PKM2 regulate the expression of several genes, including GPR174, IL13RA2, and KDM1B (Figure 6), as well as LINC00273 [9].

Our results reveal that p-Ser293 in PDHA1 is mainly associated with the C-terminal domain (aa 389–435: sky blue colour in Figure 1A) and slightly with the A1 and A2 domains (aa 44–116 and 218–388: pink colour in Figure 1A) of PKM2 [9]. We propose that p-PDHA1 directly binds to PKM2 and prevents PKM2 from ubiquitination and degradation, similar to the role of follistatin-like protein 1 (FSTL1) in reducing PKM2 ubiquitination. Our findings are consistent with previous studies that showed that FSTL1 knockout mouse reduced PKM2 levels in the liver [15]. Furthermore, MG132 and leupeptin protected PKM2 from degradation even in the presence of si-PDHA1, further supporting the idea that p-PDHA1 protects PKM2 from degradation (Figure 3D). Additionally, we identified acetylated K305 in the A2 domain of PKM2 as critical for PKM2 stability in the absence of PDHA1 (Figure 4E).

Although PKM2 has been reported to have protein kinase enzyme activity [11], we found that PKM2 could not phosphorylate PDHA1 in a complex. However, PDHA1 itself could be phosphorylated by ATP and PEP (Figure 4D,E). It would be very intriguing if PDHA1 could autophosphorylate at Ser293 by using ATP or PEP. However, the possibility that a certain protein kinase may have been contaminated in the reactants cannot be excluded.

We also found that insulin upregulated RhoA-GTP levels (Figure 5A), which increased PDHA1 Ser293 phosphorylation in HepG2 cell [8]. This suggest that RhoA-GTP may indirectly control target genes of the p-PDHA1/PKM2 complex. We focused on these target genes because they are known to have important physiological roles in various cell types and for their relevance to tumorigenesis. GPR174 is strongly expressed in melanoma cells [16], and although insulin increased GPR174 expression, it did not play a role in regulating cell proliferation. However, we cannot exclude the possibility that GPR174 may contribute to tumorigenesis through other mechanisms in response to insulin. Specifically, KDM1B, also referred to as lysine-specific demethylase 2 (LSD2), promotes the proliferation of MDA-MB-231 breast cancer cells and induces the expression of pluripotent stem cell markers NANOG and SOX2 [17]. In addition, KDM1B represses p53 expression, leading to enhanced proliferation and inhibiting apoptosis in colorectal cancer [18]. The knockdown of KDM1B has been shown to decrease breast cancer cell colony formation by increasing H3K4 methylation [19]. Through genomewide mapping, it has been revealed that KDM1B associates predominantly with the gene sets involved in active transcription. Interestingly, KDM1B forms active complexes with methyltransferase G9a and NSD3, which are both involved in transcription elongation [20]. NPAC/GLYR1 acts as a specific cofactor of KDM1B, stimulating demethylation of H3K4me1 and H3K4me2 [21].

In patients with metabolic disease such as hyperinsulinemia, including obesity, type 2 diabetes, and metabolic syndrome, the incidence of several types of cancer is increased. Interestingly, several types of cancer exhibit elevated levels of the insulin receptor [22]. That study revealed that high glucose stimulated the phosphorylation of PDHA1 at the Ser293 residue through PDK1 and PDK3 [23], suggesting that it may also trigger the formation of a p-PDHA1/PKM2 complex, impacting gene regulation. Although not all target genes of p-PDHA1 and PKM2 were studied in this work, our findings suggest that the p-PDHA1 and PKM2 complex binds to promoters of multiple specific genes and induces tumorigenesis by inducing the expression of these genes upon insulin stimulation. However, it is important to note that this study has limitations in terms of the applicability to in vivo systems, as it utilized hepatocyte carcinoma cell lines instead of primary hepatocytes. Additionally, it would be intriguing to explore the role of the p-PDHA/PKM2 complex in response to insulin in other relevant tissues such as adipocytes and skeletal muscle. Further research is necessary to investigate the regulation of gene expression in hepatocytes in response to insulin. We hypothesize that insulin may regulate histone methylation through KDM1B, which could contribute significantly to tumorigenesis in response to insulin. In conclusion, we propose that insulin enhances tumorigenesis, at least in part, through the p-PDHA1 and PKM2 complex, which functions as a transcriptional regulator.

## 4. Materials and Methods

### 4.1. Materials

Chemical-grade human recombinant insulin (Cat: INSL16-N-5) was obtained from Alpha Diagnostic International (ADI) (San Antonio, TX, USA). Recombinant Tat-C3 protein (Tat-peptide conjugated with C3 toxin) was purified by using *E. coli* BL-21 [24,25]. The Y27632 compound was collected from Millipore-Sigma (Burlington, VT, USA). The Poly-L-lysine solution (P8920), dichloroacetic acid (DCA, SB-415286), isopropyl β-D-1-thiogalctopyranoside (IPTG), and phosphoenolpyruvate (PEP) were purchased from Sigma-Aldrich Co. (St. Louis, MO, USA). The protease and phosphatase inhibitor cocktail were purchased from ApexBio (Boston, MA, USA). Dulbecco’s modified Eagle’s medium (DMEM and DMED-F12), foetal bovine serum (FBS), penicillin-streptomycin antibiotics were obtained from GibcoBRL (New York, NY, USA). Alexa flour 488 goat antimouse IgG and 4′6-diamidino-2-phenylindole (DAPI) were bought from Invitrogen (Carlsbad, CA, USA). The mounting solution, Alexa flour-568, and Alexa flour-594 secondary antibodies were obtained from molecular probes (Eugene, OR, USA). Glutathione (GSH)-Sepharose 4B agarose and protein A/G-agarose beads were obtained from Amersham Biosciences (Piscataway, NJ, USA). Antirabbit/mouse normal IgG (sc:2025 and 2729S) antibodies were purchased from Santa Cruz and Cell Signaling Technology, respectively. Anti-β-actin (sc:58673) anti-PDHA1 (sc:377092) antibodies were obtained from Santa Cruz Biotechnology (Santa Cruz, CA, USA). Anti-p-PDHA1 (Ser293, ab92696), PDH E2 (ab126224) lipoamide dehydrogenase E3 (PDH E3, ab133551) antibodies were acquired from Abcam (Cambridge, MA, USA). The anti-PKM2 (4053, JM001 and 32054) antibody was purchased from Cell Signaling Technology and Signalway, respectively. HA-probes (3724 and A02040) were obtained from Cell Signaling Technology (Danvers, MA, USA) and Abbkine (Seoul, Republic of Korea), respectively. Anti-IL-13Rα2 (ABP54898) and anti-GPR174 (ABP56771) antibodies were collected from Abbkine (Atlanta, GA, USA). Anti-KDM1B (LSC-356305/192248) antibodies were obtained from LSBio LifeSpan BioSciences, Inc. (Seattle, USA). Goat antirabbit/mouse IgG conjugated with HRP was obtained from Enzo Life Sciences (Farmingdale, NY, USA). Finally, the polyvinylidene difluoride (PVDF) membrane was obtained from Millipore (Billerica, MA, USA). Female rat liver tissues were obtained and used for the Western blot analysis depending on previous literature [9]. Formalin-fixed liver cancer tissues with corresponding normal tissues (#A304 I) were directly purchased from ISU ABXIS CO., LTD (Bundang, Republic of Korea).

### 4.2. Molecular Docking Study between p-PDHA1 and PKM2

The three-dimensional structures of PKM2 (PDB id: 1T5A) and PDHA1 (PDB id: 3EXE) were obtained from the Protein Data Bank [26,27]. The structures were prepared using the Prepare Protein module of Discovery Studio (DS) v2019 software. Since the phosphorylated structure of PDHA1 at Ser264 (p-PDHA1) was not available in PDB, a phosphorylation addition to the structure was performed using the Build and Edit Protein module of DS. The structures were then energy-minimized using the Steepest Descent algorithm and saved in PDB format. Subsequently, the minimized structures of the p-PDHA1 and PKM2 were submitted to the HADDOCK2.4 and ClusPro web servers to perform protein–protein docking with the default parameters [28,29]. HADDOCK2.4 is a docking method that uses a data-driven approach for docking with the support of experimental knowledge [30], while ClusPro performs rigid-body docking by sampling billions of conformations. The server uses the PIPER program based on the fast Fourier transform (FFT) approach, utilizing geometric matching [31]. During the docking experiment, PKM2 was used as a receptor, and p-PDHA1 was used as a ligand. Our experimental studies identified phosphorylated Ser264 of PDHA1 forming interactions with the C-terminal domain of PKM2, which is the allosteric site of this protein. This information was used in the HADDOCK2.4 knowledge-based docking experiment to define the interaction site between the proteins. In contrast, the docking experiment with ClusPro was run without providing the information of interacting residues. The docking results were analysed in DS, and the final model was selected based on various statistical parameters, cluster analysis, and fundamental residual interactions [32,33].

### 4.3. Cell Culture

The cell lines used in this study were obtained from Korean Cell Line bank (Seoul, Republic of Korea) and maintained in Dulbecco’s modified Eagle’s medium (DMEM) either supplemented with 10% heat-inactivated foetal bovine serum (FBS) or 1% penicillin-streptomycin antibiotics (100 units/mL of penicillin-streptomycin). After that, all human hepatocellular carcinoma (HCC) cells were maintained in a humidified atmosphere with 5% CO_2_ and 95% air at 37 °C in an incubator for proper growth [8,9].

### 4.4. Western Blot Analysis

After the completion of the treatment period, the cells were washed with 1× PBS and harvested for further analysis. Cell lysis was performed with a RIPA buffer (50 mM Tris HCl pH 7.5, 1 mM MgCl_2_, 1% Nonidet P40, 150 mM NaCl) supplemented with freshly prepared phosphatase/protease cocktail inhibitors (ApexBio). The cell lysates were then centrifuged at 13,000× *g* for 20 min at 4 °C. The protein concentration in the resulting supernatant was quantified by the BCA method. SDS-PAGE was performed using protein samples (20 to 30 mg/lane), followed by transfer to PVDF membranes. The blots were probed with target antibodies and processed using previous established protocols [34].

### 4.5. Immunoprecipitation (Co-IP)

After treatment, approximately 1 × 10^7^ cells were washed with 1× PBS and lysed in a RIPA buffer (20 mM Tris pH 7.4, 120 mM NaCl, 1 mM MgCl_2_, 1% Nonidet P-40) supplemented with freshly prepared 1% of protease/phosphatase inhibitor cocktail (ApexBio). The cell lysates were then cleared by centrifugation at 13,000× *g* for 20 min at 4 °C. The cleared supernatants were precleaned by incubating with protein A/G-agarose beads for 1 h and then incubated again with anti-IgG or anti-p-PDHA1 and PKM2 (1:100–1:500 dilution) antibodies for 3 h at 4 °C. Finally, protein A/G agarose beads (30 µL) were then added to the lysates and incubated with shaking at 4 °C for 3 h, followed by previous protocols [35].

### 4.6. GTP-RhoA Pull-Down Assay

For GTPγS or GDP-binding to protein in vitro as controls, 10 µL of EDTA (0.5 M pH 8.0) was added to 500 µL cell lysates containing 500 µg of protein to achieve a final concentration of 10 mM. Next, 5 µL of 10 mM GTPγS (for a final concentration of 0.1 mM) or 5 µL of 100 mM GDP (for a final concentration of 1 mM) was added to the above cell lysate and mixed with a vortex mixer. Then, mixtures were then incubated at 30 °C for 30 min. To terminate the reaction, the samples were placed on ice and 32 µL of 1 M MgCl_2_ (for a final concentration of 60 mM) was added (Pierce). For the GTP-RhoA pull-down assays, cells were starved serum-free DMEM for 12 h before the assays. The cells were then stimulated and washed with ice-cold 1× PBS and lysed in lysis buffer A (25 mM Tris pH 7.5, 150 mM NaCl, 1% Nonidet P-40, 5 mM MgCl_2_, 5% glycerol, and protease inhibitors including 1 mM PMSF, 1 μg/mL each of leupeptin, aprotinin, and pepstatin A). The lysates were clarified by centrifugation and equalized for total volume and protein concentration. The lysates were incubated with GST-Rhotekin-Rho binding domain (RBD) beads, washed three times with ice-cold lysis buffer B (50 mM HEPES pH 7.4, 150 mM NaCl, 1% Triton X-100, 0.5 mM MgCl_2_, and 100 µM Na3VO_4_ with protease inhibitors). The bound fraction (active RhoA-GTP) was separated on SDS-PAGE [36]. Active RhoA and total RhoA were analysed by Western blotting with an anti-RhoA antibody (monoclonal antibody 26C4; Santa Cruz). The results were quantified using Photoshop 7.01 software (Adobe, San Jose, CA, USA) and statistical significance was determined using Prism 4 software (GraphPad, La Jolla, CA, USA).

### 4.7. In Vitro Kinase Assay

GST-PDHA1 was purified and treated with alkaline phosphatase (ALP, 3 U/μL) and DCA (20 mM) in an ALP reaction buffer (10 mM Tris-HCL, 20 mM NaCl, 5 mM MgCl_2_, 0.1 mM MnCl_2_, and 0.1 mM DTT) for 1 h to dephosphorylate the GST-PDHA1 protein. Subsequently, GST-PDHA1 was washed with a kinase assay buffer and then 0.2 µg of GST-PDHA1 protein was incubated with the recombinant PKM2 protein (0.2 µg) in the presence and absence of ATP or 30 μM PEP in a kinase assay buffer (20 mM HEPES pH 7.5, 2 mM β-glycerophosphate, 20 mM MgCl_2_, and 1 mM EDTA) for 30 min at 25 °C. After incubation, the samples were washed three times with kinase assay buffers and the supernatants were discarded. Finally, the samples were analysed with a Western blot analysis.

### 4.8. GST Fusion Protein Purification

The target genes (GST-PDHA1 and GST-PKM2) were cloned into *E. coli* BL-21 competent cells for protein expression. The expression of fusion proteins was induced by IPTG (0.3 to 0.8 mM) for 2 to 3 h at room temperature with agitation (80–100 rpm). The bacterial cells were lysed with a lysis buffer which contained 50 mM HEPES (pH 7.4), 150 mM NaCl, 5 mM MgCl_2_, 1% TX-100, 0.01% SDS, and protease inhibitors were freshly added, including 1 mM PMSF, 1 μg/mL each of leupeptin, aprotinin, and pepstatin A plus phosphatase inhibitors (1 mM NaF and Na_3_VO_4_). The lysates were cleared by centrifugation at 13,000× *g* and incubated with GSH-conjugated agarose resin at 4 °C overnight on a shaker. The resins were washed three times with a lysis buffer without detergent. In some experiments, GST was removed by using thrombin (Sigma), and the thrombin was cleared by p-aminobenzamidine-agarose (Sigma, A7155). Finally, the GST-fusion protein was stored in a buffer (50 mM HEPES pH 7.4, 100 mM NaCl, 10% glycerol, and 1 mM DTT) and kept in aliquots at −70 °C for further use.

### 4.9. Cell Proliferation Assay

The cells were initially seeded in either 12-well dishes at a density of 1 × 10^5^ cells/well or in 6-well dishes at a density of 4 × 10^5^ cells/well. Then, cells were serum-starved for 12 h before being treated with 100 nM insulin. Following the treatment, the media was removed, and the cells were washed twice with 1× PBS. Next, 10 μL of CCK-8 reagent (highly water-soluble tetrazolium salt-WST-8 (2-(2-methoxy-4-nitrophenyl-3-(4-nitrophenyl0-5-(2,4-disulfophenyl)-2H-tetrazolium, monosodium salt) in 100 μL of serum free medium was added to each well, and cells were incubated at 37 °C in a 5% CO_2_ atmosphere for 1–4 h. The optical density values were then measured at 450 nm in a spectrophotometer. To ensure reproducibility, the MTT assays were repeated three times in every experiment.

### 4.10. Plasmid DNA and si-RNA Transfection

In all transfection experiment, cells were initially seeded at 50–60% confluency in a six-well plate and incubated for 8 h in growth media. The cells were transfected using the jetPRIME transfection reagent (Polyplus transfection, France) according to the manufacturer’s protocols. Small interfering RNAs (si-RNA) PDHA1 (sc-91064 and 5160), si-PKM2 (sc62820 and 5315), and control si-RNA-A (sc-37007) were purchased from Santa Cruz and Bioneer. Si-RNA GPR174 (84636-2), si-RNA Il13Ra2 (3598-1), si-RNA KDM1B (221656), and si-IR-A (customized sequence; sense strand 5′ CUAGUCCUGC-AGAGGAUUU- 3′ and antisense strand 5-′ AAAUCCUCUGCAGGACUAG- 3′) [37] were also bought from Bioneer (Daejeon, Republic of Korea). These si-RNAs were transfected at a final concentration of 30–100 nM, and 2–3 µg DNA was transfected to the cells. Briefly, si-RNA and DNA were diluted into in 200 µL of the jetPRIME buffer, then 2–9 µL of the jetPRIME reagent was added, and the mixture was incubated for 10 min. Finally, the solutions were added to the six-well plate and incubated for another 24–48 h before performing the stated experiments.

### 4.11. Site-Directed Mutagenesis

Flag-PCDH-CMV PKM2-K305Q, PKM2R, HA-pcDNA3.1 PDHA1 S293D, and S293A constructs were generated using a site-directed mutagenesis kit (Intron Biotechnology, Sungnam, Republic of Korea) with the primer designed with specific mutations, following the manufacturer’s protocol [34]. pCMV6-XL5 PDHA1 (human cDNA clone) was obtained from OriGene Technologies, Inc. (Rockville, MD, USA), and the PCDH-CMV PKM2 construct was kindly provided by Dr Seong-Hoon, Park.

### 4.12. Pyruvate Kinase M2 Activity Assay

A pyruvate kinase activity colorimetric assay kit (K709-100, BioVision, Milpitas, CA, USA) was used to measure the pyruvate kinase activity in HepG2 cells stimulated with insulin (100 nM) for 30 min. Cells were ruptured with 4 volumes of the PK assay buffer (BioVision) with a homogenizer (Forte 100, Saeshin, Daegu, Korea), followed by the removal of cell debris by centrifugation at 15,000× *g* for 10 min. The resulting cell lysates were then used to convert phosphoenolpyruvate (PEP) plus ADP to pyruvate plus ATP. The generated pyruvate is oxidised by pyruvate oxidase, which produce a colour that was measured by reading optical density values at 570 nm using an optical density plate reader or a spectrophotometer (SpectraMax Plus 384, Molecular Devices, San Jose, CA, USA). The reaction incubation time was 10–30 min at 25 °C.

### 4.13. PDH Activity Assay

HepG2 cells were cultured in growth media for 24 h. Following this, the cells were washed with PBS (1.4 mM KH_2_PO_4_, 8 mM Na_2_HPO_4_, 140 mM NaCl, and 2.7 mM KCl, pH 7.3) and incubated in DMEM without serum for 12 h before treatment. The cells were harvested, and the cytosolic protein lysates of HepG2 cells were prepared by adding a lysis buffer (50 mM HEPES (pH 7.4), 0.5% TX-100, 0.5 mM EDTA, 1 mM DTT, 1 mM PMSF, and protease inhibitors cocktail) to the cells. The protein concentration was determined using a BCA assay kit (Thermo Fischer Scientific, Waltham, MA, USA). PDH activity was measured using a colorimetric assay kit (TB561, Novagen). Briefly, 200 μL of cytosolic lysates (2.5 μg/μL) was added to the wells of a microplate precoated with anti-PDH antibody and incubated for 1 h at room temperature. The unbound proteins were removed by washing with the assay buffer (25 mM Tris-HCl, pH 7.5, 0.5 mM EDTA, 0.5% Triton X-100). The PDH activity was measured by following the reduction of NAD^+^ to NADH, coupled with the reduction of a reporter dye to yield a yellow-coloured reaction product, which was monitored by measuring the increase in absorbance at 450 nm using a spectrophotometer (SpectraMax Plus 384, Molecular Devices). The PDH activity was expressed as the rate of NADH formation per minute per milligram of protein.

### 4.14. Fluorescence Microscopy Assay of Liver Cancer Tissues and Cells

HepG2 cells were seeded onto glass cover slips and incubated for 16 h. Then, cells were serum-starved for 12 h and stimulated with insulin for a specific amount of time, fixed with paraformaldehyde. After permeabilization with 0.2% Triton X-100, the cells were incubated in Tris/NaCl buffer containing 0.5% BSA for 30 min. For staining with rabbit polyclonal antibodies, the cells were incubated either with a 1:100 dilution of anti p-PDHA1 or anti-PKM2 antibody in the blocking solutions at room temperature for 1 h, washed three times with a Tris/NaCl buffer containing 0.1% Triton X-100, and then stained with Alexa-546-labelled goat antirabbit IgG (Amersham). For the dual immunofluorescence, either monoclonal anti-p-PDHA1 or monoclonal anti-PKM2 at a 1:50 dilution was added and incubated with the cells overnight at 4 °C. Rhodamine-conjugated antirabbit IgG and Alexa-488-conjugated antimouse IgG (Molecular Probes) with a 1:50 dilution were used for the dual immunofluorescence staining, and the nuclear region was stained with DAPI (4′,6-diamidino-2-phenylindole).

For formalin-fixed liver cancer tissues, the slide was de-paraffinized and rehydrated with xylene and ethanol solutions. Antigen retrieval was performed using 10× sodium citrate buffer (pH 6) at 100 °C for 30 min. After washing with 1× PBS, the slide was incubated with the antibody in 1× PBS overnight at 4 °C and then washed with 1× PBS. The cells and tissues were stained with Alexa-546 (red) or Alexa-488 (green) labelled goat anti-rabbit IgG, and the nuclear region was stained with DAPI. Fluorescence images were collected with a conventional fluorescence microscope (Axiovert 200, Zeiss, Oberkochen, Germany) [38].

### 4.15. Chromatin Immunoprecipitation (ChIP) Sequencing

HepG2 cells were cultured in 100 mm (diameter) dishes at a density of 1 × 10^7^–3 × 10^7^ cells per dish and incubated for 24 h. The cells were then serum-starved for 12 h and stimulated with insulin (100 nM) for 48 h. After washing with 1× PBS, the cells were cross-linked with formaldehyde dropwise directly to the 1× PBS to a final concentration of 0.75% and incubated at room temperature for 1 h. Crosslinking was stopped by adding glycine to a final concentration of 125 mM and incubated again for 10–15 min at room temperature. The cells were rinsed twice with ice-cold PBS and harvested by centrifuge at 13,000 rpm for 10 min at 4 °C. The pellets were suspended with a ChIP lysis buffer (50 mM Tris-HCl, pH 7.4, 1% NP-40, 0.25% Na-Deoxycholate, 150 mM NaCl, 1 mM EDTA, 0.1% SDS, 0.5 mM DTT, 5 mM sodium butyrate) freshly supplemented with proteases/phosphatases inhibitors cocktail and incubated for 10 min on ice. The lysates were sonicated using a Covaris sonicator to obtain a chromatin fragment of 200–500 bp. The chromatin was then incubated with antibodies against the protein of interest at 4 °C overnight with a gentle rotation. The immunocomplexes were captured using protein A/G beads and washed with low and high salt buffers, LiCl buffer, and TE buffer, successively. Finally, the bound chromatin was eluted and reverse-cross-linked to obtain purified DNA [9]. ChIP sequencing was conducted by Ebiogen Inc. (Seoul, Republic of Korea) using the Illumina sequencing platform.

### 4.16. Statistical Analysis

ImageJ 1.53 and Photoshop CC2018 (Adobe, San Jose, CA, USA) were used for the statistical analysis of the Western blotting band intensity and imaging, respectively, while PRISM 4.0 and 8.0 software (GraphPad, San Diego, CA, USA) were used for the data analysis and graphing. In general, the results are shown as mean ± SE of at least three independent experiments, and the Western blotting protein bands are representative of several independent experiments. The statistical analysis of significance was based on Student’s *t*-test and a 1- or 2-way ANOVA, and the level of significance was determined based on *p*-values. Specifically, * *p* < 0.05 was considered significant, ** *p* < 0.01 more significant, and *** *p* < 0.001 being highly significant.

## Figures and Tables

**Figure 1 ijms-24-13697-f001:**
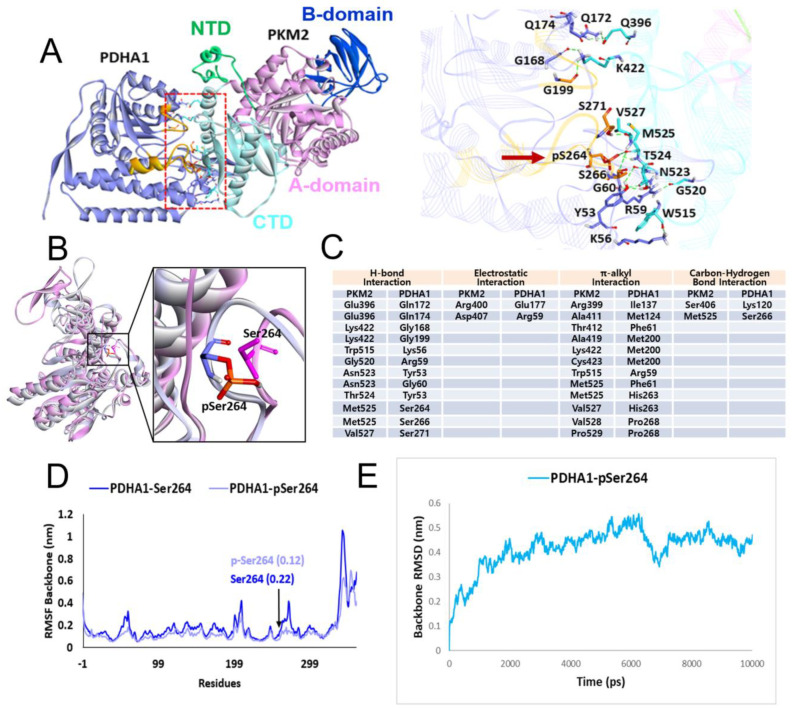
**In silico analysis of interaction between p-PDHA1 and PKM2**. (**A**) HADDOK2.4 provided the statistic of 10 clusters from 164 possible complexes. Among them, the most favourable model, which superimposed with other models, aligned perfectly, and occupied similar binding regions, was selected. Conformations of binding proteins of p-PDHA1 (violet) and PKM2 (green: N-terminal domain, pink: A1 and A2 domains; blue: B domain; and sky blue: C-terminal domain) are shown (left panel). Detailed amino acids residues involved in the protein interaction are revealed, and the p-Ser264 residue is shown in arrow (right panel). (**B**) The detailed conformations of Ser264 (pink) and p-Ser264 (grey) are shown. (**C**) Amino acids participating in interaction between p-PDHA1 and PKM2 with H-bond, electrostatic, π-alky, and carbon–hydrogen bonds are summarized. (**D**) The RMSF (root-mean-square fluctuation), which indicates specific amino acid structurally contributing the most to molecular motion, is shown. (**E**) The RMSD (root-mean-square deviation), which means a numeric measurement representing the difference between two structures is presented.

**Figure 2 ijms-24-13697-f002:**
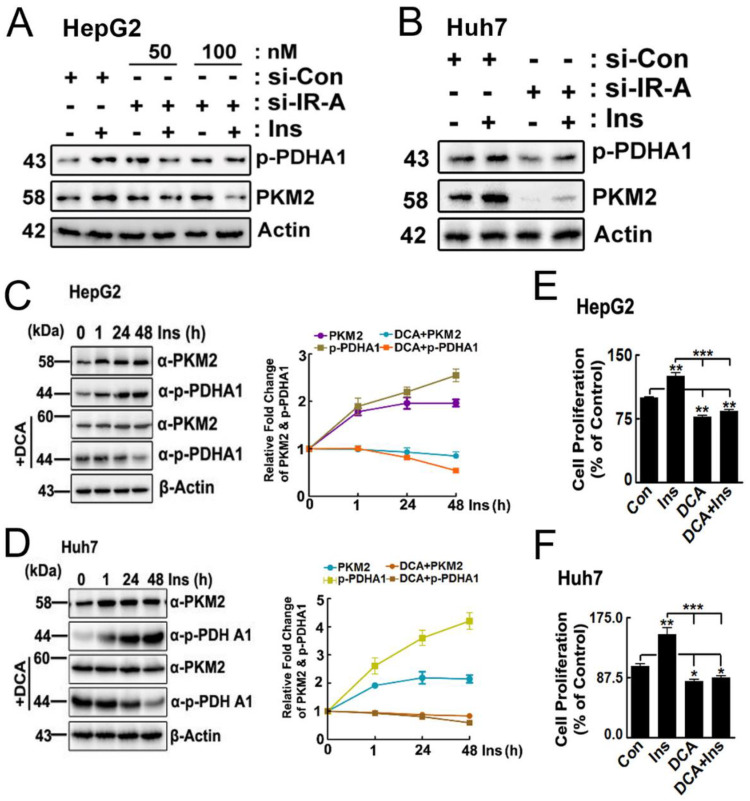
**DCA reduces the cell proliferation and p-PDHA1 level upon insulin simulation**. (**A**,**B**) HepG2 and Huh7 cells were transfected with si-IR-A and stimulated with insulin (100 nM) for 24 h. p-PDHA1 and PKM2 levels were determined by Western blotting. (**C**,**D**) HepG2 and Huh7 cells were stimulated with 100 nM insulin in the absence or presence of DCA (25 mM). PKM2 and p-PDHA1 levels were detected by Western blotting. Statistical data are presented in a line graph. (**E**,**F**) Human HepG2 and Huh7 cells were stimulated with 100 nM insulin, 25 mM DCA, with or without insulin for 48 h, and finally, cell proliferation was measured by using an MTT assay. The relative growth was quantified as a percentage of control cells. Data are presented in a bar graph and the data represent the mean ± S.E. of three independent experiments (* *p* < 0.05; ** *p* < 0.01; *** *p* < 0.001).

**Figure 3 ijms-24-13697-f003:**
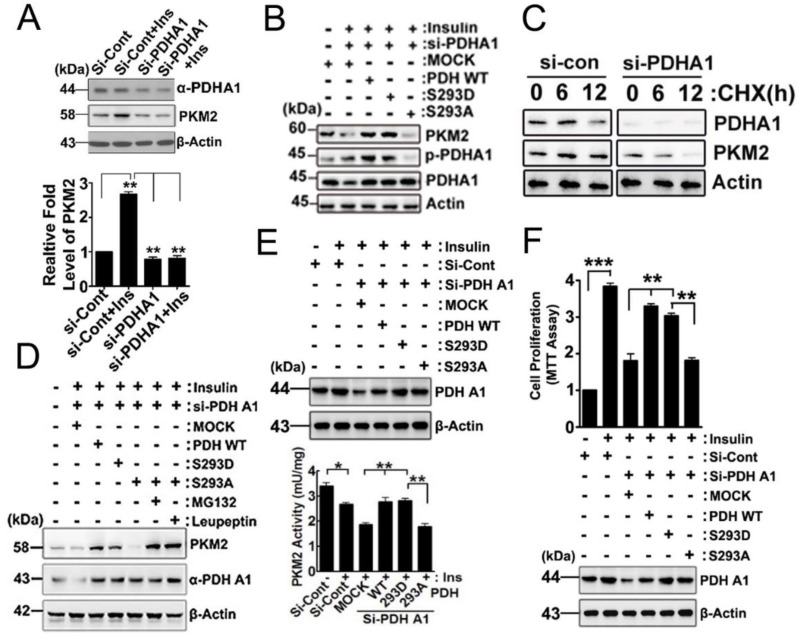
**Effects of p-PDHA1 on PKM2 upon insulin stimulation**. (**A**) HepG2 cells were transfected with control si-RNA or si-PDHA1 (30–100 nM) for 48 h, and PKM2 and p-PDHA1 levels in transfected HepG2 cells were determined by Western blotting. PKM2 levels were quantified and are presented as the mean ± S.E. of three independent experiments (** *p* < 0.01). (**B**) HepG2 cells were transfected with si-PDHA1 and reconstituted with PDHA1 WT, S293D, or S293A, then activated by insulin (100 nM) for 48 h. Target proteins band were determined by Western blotting. (**C**) si-PDHA1 (50 nM) was transfected to HepG2 cells, cyclohexamide (CHX) (20 mM) and insulin (100 nM) were administered for a defined time, then p-PDHA1 and PKM2 were determined with Western blotting. (**D**) HepG2 cells transfected with si-PDHA1 and reconstituted with PDHA1 S293A were stimulated with 100 nM insulin in the presence of MG132 (1 μM) or leupeptin (20 μM). PDHA1 and PKM2 level changes were determined by Western blotting. (**E**) HepG2 cells were transfected with si-PDHA1 and reconstituted with PDHA1 WT, S293D, or S293A mutants. Then, cells were stimulated by 100 nM insulin, and PKM2 enzyme activity levels were measured by using the BioVision enzyme colorimetric activity assay kit (K709-100, BioVision, Milpitas, CA, USA). (**F**) HepG2 cells were transfected first with si-PDHA1 and reconstituted with PDHA1 WT, S293D, or S293A, then cells were activated with insulin (100 nM) for 48 h. Cells proliferation were measured by a CCK-8 assay. The levels of knockdown or reconstitution of target constructs were checked by Western blotting. The data represent the mean ± S.E. of three independent experiments (* *p* < 0.05, ** *p* < 0.01; *** *p* < 0.001).

**Figure 4 ijms-24-13697-f004:**
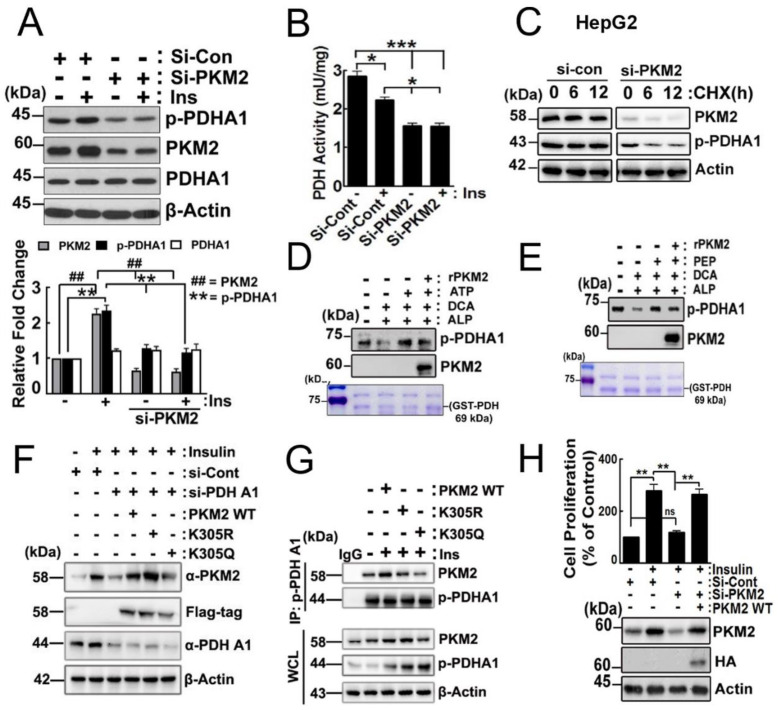
**Effects of PKM2 on p-PDHA1 upon insulin stimulation**. (**A**) HepG2 cells were transfected with si-PKM2 and p-PDHA1, and PDHA1 levels were measured by immunoblotting after a 100 nM insulin treatment for 48 h. (**B**) Si-PDHA1 (50 nM) was transfected and cyclohexamide (CHX) (20 mM) and insulin (100 nM) were administered; then, p-PDHA1 and PKM2 levels were determined with Western blotting depending on time. (**C**) HepG2 cells were transfected with si-PKM2 and stimulated by 100 nM insulin. PDH enzyme activity was then measured by a PDH activity colorimetric assay protocol (Novagen, Darmstadt, Germany). (**D**,**E**) Purified recombinant GST-PDHA1 was first treated with alkaline phosphatase to remove endogenous phospho-groups in PDHA1. GST-PDHA1 was then incubated with recombinant PKM2 (0.2 μg) and PEP (20 μM) or ATP (30 μM), and p-Ser293 PDHA1 was detected by Western blotting. Purified GST-PDH-WT protein band was stained with Coomassie blue. (**F**) HepG2 cells were transfected with si-PDHA1 and then PKM2 K305R and K305Q mutants, and cells were stimulated with 100 nM insulin. The PDHA1 and PKM2 levels were determined by Western blotting. (**G**) HepG2 cells were transfected with PKM2 WT, K305R, and K305Q and then stimulated with insulin (100 nM). P-PDHA1 immunoprecipitated and coimmunoprecipitated PKM2 was detected by Western blotting. (**H**) HepG2 cells were transfected with si-PKM2 and reconstituted with PKM-WT and stimulated by insulin (100 nM) for 48 h. Cell proliferation was assessed with a CCK-8 assay, and PKM2 levels were determined by Western blotting. The statistical data represent the mean ± S.E. of three independent experiments (* *p* < 0.05, ** *p* < 0.01; *** *p* < 0.001, ns, non-significant).

**Figure 5 ijms-24-13697-f005:**
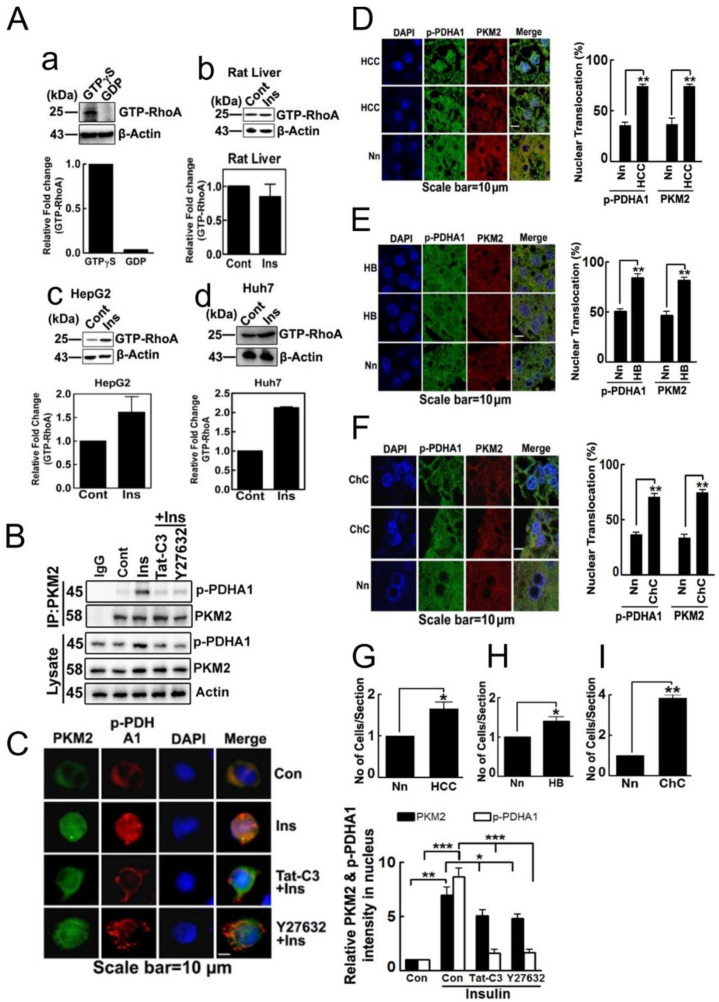
**Nuclear localization of p-PDHA1 and PKM2 in human hepatocellular carcinoma cells and tissues**. (**A**) Control experiment of pull-down assay of RhoA-GTP using RBD-bead showed that GTPγS-bound RhoA but not RhoA-GDP preferentially interacted with RBD-beads (a). Then, RhoA-GTP levels were measured in normal rat liver tissue, stimulated by insulin (100 nM) for 3 h (b). Finally, RhoA-GTP levels were measured in HepG2 (c) or Huh7 cells (d), which were stimulated by 100 nM insulin for 1 h. (**B**) HepG2 cell were pretreated with Tat-C3 (2 μg) or Y27632 (10 μM) for 2 h and then treated with insulin (100 nM). PKM2 immunoprecipitated and coimmunoprecipitated p-PDHA1 was determined with Western blotting. (**C**) HepG2 cells were pretreated with Tat-C3 (2 μg) or Y27632 (10 μM) for 2 h, and then cells were activated with 100 nM insulin for 48 h; p-PDHA1 and PKM2 species were identified by immunohistochemistry. P-PDHA1 and PKM2 fluorescence intensities were determined and are presented as the mean ± S.E. of three independent experiments (* *p* < 0.05, ** *p* < 0.01, *** *p* < 0.001). (**D**–**F**) Hepatocellular carcinoma (HCC) (**D**), hepatoblastoma (HB) (**E**), cholangiocarcinoma (ChC) (**F**) and non-neoplastic (Nn) tissue samples were immunostained with p-PDHA1 and PKM2 antibodies (green and red fluorescence, respectively) and the cells’ nuclei were stained with DAPI (blue fluorescence). Relative intensities of p-PDHA1 in HCC, HB, ChC, and Nn were quantified, and the relative nuclear levels were also quantified. (**G**–**I**) Cell numbers were counted in HCC (**G**), HB (**H**), and ChC (**I**). The statistical data represent the mean ± S.E. of three independent experiments (* *p* < 0.05, ** *p* < 0.01 and *** *p* < 0.001).

**Figure 6 ijms-24-13697-f006:**
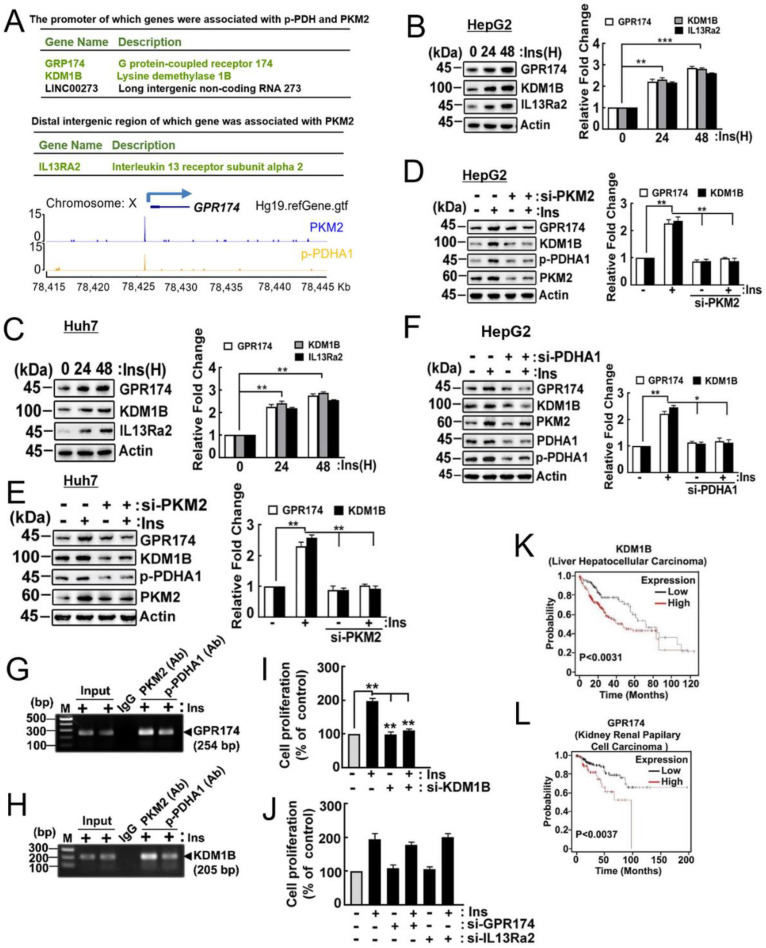
**Target genes of p-PDHA1 and PKM2**. (**A**) ChIP sequencing analysis was performed using p-PDHA1 and PKM2 antibodies. The promoters of GP174 and KDMB1 were associated with p-PDHA1 and PKM2 antibodies, and the distal intergenic region of IR13A2 was associated with PKM2 antibody. The peaks of p-PDHA1 and PKM2 antibodies are presented. (**B**,**C**) HepG2 or Huh7 cells were treated with insulin (100 nM) at the indicated time, and target genes KDM1B, GPR174, and IL-13Rα2 protein levels were determined by Western blotting. (**D**,**E**) Human HepG2 and HuH7 cells transfected with si-PKM2 cells were activated by insulin (100 nM) for 24 h. Finally, the target proteins were measured by Western blotting. (**F**) HepG2 cells were transfected with si-PDHA1, and then cells were stimulated with insulin (100 nM) for 24 h. Then, the indicated protein levels were determined by Western blotting. (**G**,**H**) HepG2 cells were first treated with 100 nM insulin, and then crosslink samples were harvested. Finally, ChIP-PCR of GPR174 and KDM1B promoter was conducted by immunoprecipitation with p-PDHA1 and PKM2 antibody, respectively. (**I**,**J**) HepG2 cells were transfected with si-KDMB1 (**I**) and si-GPR174 and si-IL13a2 for 24 h and stimulated with insulin (100 nM) for 48 h. Cell proliferation was determined by an MTT assay. (**K**,**L**) The survival expression of KDM1B and GPR174 was obtained and plotted via a Kaplan–Meier plotter (Pan-Cancer RNAseq, TCGA survival). The statistical data are presented in a bar diagram; the data represent the mean ± S.E. of three independent experiments (* *p* < 0.05, ** *p* < 0.01, and *** *p* < 0.001).

## Data Availability

Data set is not available.

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
