# Peer review of "Association of Phosphorylated Pyruvate Dehydrogenase with Pyruvate Kinase M2 Promotes PKM2 Stability in Response to Insulin"

_ijms, 2023, doi:10.3390/ijms241813697_

Round 1
Reviewer 1 Report
The authors present a thorough analysis of the interplay and dynamic of PDHA1 serine phosphorylation and PKM2 levels based on previous findings. Overall the manuscript is well written and presents data well. A few points to consider:
-Although not the aim of your study, can you surmise in the discussion what the presence of glucose or glucotoxic conditions in your model would have on the associations you observed?
-can you further detail exactly how you identified your groups of target genes through CHIP sequencing? Was there a particular cutoff you used for expression/abundance etc?
-Line 100, i think you mean figure 2a and 2b
-I think it is useful to have a few sentences on limitations of the current work and future directions. Do you plan to investigate other metabolically relevant tissues (adipose, skeletal muscle, etc)?
Author Response
Thank your comprehensive and constructive comments. We attached the response to reviewer 1 comment.

Reviewer 2 Report
Dear authors
I read with interest the manuscript entitled “Association of phosphorylated pyruvate dehydrogenase with pyruvate kinase M2 promotes PKM2 stability in response to insulin”
Herein, you can find my comments.
This study deals to clarify molecular mechansim that contribute to stabilizzazion of PKM2 after insulin stimulation.
The authors demonstrated that insulin promotes the phosphorylation of PDHA1 in Ser293 and the phosphorylated form of this protein is able to recruit PKM2, avoiding its degradation through a proteasome-dependent mechanism. The formation of the PDHA1-PKM2 complex has a key role in promoting the proliferation of liver cancer cells. Interestingly, although binding of PKM2 to p-PDHA1 has been shown to regulate PDHA1 activity, this effect is not due to the ability of PKM2 to phosphorylate PDHA1. Finally, PKM2 is destabilized upon acetylation of the Lys305 residue, and interaction of the acetylated form of PKM2 with PDHA1 prevents PKM2 degradation. Further analyzes show that insulin led to the upregulation of PKM2 and p-PDHA1 levels and their nuclear localization. Analysis of patients with liver tumors confirmed that nuclear levels of PDHA1 and PKM2 are elevated, suggesting that nuclear localization of p-PDHA1 and PKM2 could be a characteristic of liver tumor tissue. Furthermore, the authors demonstrated that the PKM2-p-PDHA1 complex plays an important role in regulating the expression of the GPR174 and KDM1B genes, which play a role in the development and progression of certain types of cancers. Overall, I find this study interesting. However, I believe this manuscript contains some flaws that should be ironed out before publication in the International Journal of Molecular Sciences
Major points
Figure 4C: The quality of the blot is poor. The authors stated, “…Furthermore, si-PKM2 treatment resulted in a significant reduction of p-PDHA1 levels in the presence of cyclohexamide (CHX), a protein synthesis inhibitor.” However, the p-related signal -PDHA1 is of bad quality: it looks like a non-specific signal rather than a band generated by the antibody response. I ask the authors to replace this image with a higher quality blot
All tests were performed using 100 nM insulin. However, at this concentration insulin is able to bind both insulin A receptor, insulin B receptor and IGF1. Both in the introduction and in the discussion no mention is made of the possible receptor that could be stimulated under the experimental conditions used. I suggest the authors clarify this point.
In most tumor cells, IR is overexpressed and the IR-A isoform of the mitogen is predominant. Can the authors confirm that the observed effects are mediated by stimulation of IR-A isoforms?
The authors demonstrated that insulin via the RhoA-GTP/PDHA1 axis indirectly regulates the expression of several genes such as GPR174, IL13RA2 and KDM1B. Furthermore, the authors hypothesized that these genes are important in promoting the proliferation of cancer cells. However, no direct evidence to support this hypothesis was presented in the results section. Therefore, the possible involvement of this gene in supporting tumor cell proliferation after insulin stimulation appears speculative. I suggest the authors to implement this part to confirm their hypothesis
Minor points
Lane 99-104
Pleae correct Fig 1A and B in Fig.2A and B.
Author Response
Thank your comprehensive and constructive comments. We attached the response to reviewer 2 comment.

Round 2
Reviewer 2 Report
Dear authors,
I read the revised version of the manuscript entitled "Association of phosphorylated pyruvate dehydrogenase with pyruvate kinase M2 promotes PKM2 stability in response to insulin"
I appreciate your efforts to improve the quality of the manuscript. I think the new version is suitable for publication in the International Journal of Molecular Sciences
Regards
Author Response
-